# The Relationship between Physical Activity and College Students’ Mobile Phone Addiction: The Chain-Based Mediating Role of Psychological Capital and Social Adaptation

**DOI:** 10.3390/ijerph19159286

**Published:** 2022-07-29

**Authors:** Hanwen Chen, Caixia Wang, Tianci Lu, Baole Tao, Yuan Gao, Jun Yan

**Affiliations:** College of Physical Education, Yangzhou University, Yangzhou 225127, China; mx120200454@stu.yzu.edu.cn (H.C.); mx120190362@stu.yzu.edu.cn (C.W.); mz120200752@stu.yzu.edu.cn (T.L.); dx120190064@yzu.edu.cn (B.T.); mx120210486@stu.yzu.edu.cn (Y.G.)

**Keywords:** physical activity, mobile phone addiction, psychological capital, social adaptation, the intermediary effect

## Abstract

The purpose of this study was to investigate the effects and mechanisms of physical activity on mobile phone addiction among college students. A total of 9406 students, ranging from freshmen to seniors, from 35 colleges in four regions of Jiangsu Province were selected using the whole group sampling method. Questionnaires, particularly the International Physical Activity Questionnaire—Long Form (IPAQ), the positive psychological capital scale (PPQ), the social adjustment diagnostic questionnaire (SAFS), and the mobile phone addiction index scale (MPAI), were administered. We found that physical activity negatively predicted mobile phone addiction among university students. Social adaptation partially mediates between physical activity and mobile phone addiction among university students, with separate mediation of psychological capital playing no indirect role. Psychological capital and social adjustment mediate the chain between physical activity and mobile phone dependence among college students. Our findings suggest that physical activity is an important external factor influencing college students’ mobile phone dependence, and it indirectly affects university students’ mobile phone addiction through psychological capital and social adaptation. Improving the physical activity level of college students, enhancing their psychological capital, and promoting improved social adaptation are important ways to prevent mobile phone addiction among college students.

## 1. Introduction

According to China’s 49th statistical report on Internet development in December 2021, 34.9% of Chinese Internet users <30 years of age used mobile phones to access the Internet [1]. Although mobile phones provide a convenient way for young people to learn and live, their convenience and accessibility also increase the risk of excessive use by individuals, negatively impacting young people [2] and leading to mobile phone addiction [3]. Mobile phone addiction has now become a social problem affecting the healthy development of young people. Studies have shown that mobile phone addiction among Chinese adolescents is rising annually, exceeding 25% [4], and the prevalence of mobile phone addiction among children and young people worldwide is 23.3%, which is also not optimistic [5]. Mobile phone addiction affects not only the physical and mental health of adolescents [6,7,8] but also their academic performance [9,10,11], leading to negative consequences such as sleep disturbances [12], hindering social relationships [13], and impairing emotional well-being in daily life [14,15].

Mobile phone addiction is defined as an individual’s excessive addiction to various activities mediated by mobile phones, resulting in a strong and persistent sense of craving and dependence on mobile phone use that leads to significant impairment of social and psychological functioning [16]. This term is also known as mobile phone dependence, problematic mobile phone use, etc., and is essentially a behavioral addiction [17]. The global rate of insufficient physical activity levels among children and adolescents is 81%, and only 13.1% of children and adolescents in China meet the recommended amount of daily physical activity [18,19]. According to the theory of temporal self-regulation of physical activity [20], many beneficial processes are triggered by physical activity. We still believe that physical activity is also an important factor influencing mobile phone addiction among young people. For example, physical exercise may have physiological benefits, whereas exercising with others or self-mastering an exercise contributes to mental health [20]. Repeated physical activity often promotes other positive health behavior changes [21]. Some research found that physical activity is a significant negative predictor of adolescent mobile phone addiction [22] and that physically inactive adolescent students are more likely to have problems with smartphone use than those who are physically active [23]. Students with a high physical activity rating who prefer physical activity had the lowest overall scores for mobile phone addiction, the lowest scores for all dimensions, and the lowest detection rates for mobile phone addiction [24,25]. It was also found that girls are more likely to have problems with smartphone use than boys [23]. Therefore, we hypothesized that physical activity negatively predicts mobile phone addiction among university students. However, the effects of physical activity on mobile phone addiction must be explored further. The mechanisms of physical activity’s effect on mobile phone addiction must also be explored from a multifactorial integration perspective. Identifying the mediating mechanisms involved is crucial to promoting the prevention of mobile phone addiction and developing interventions for adolescents.

Psychological capital may have an important role in performing between physical activity and college students’ mobile phone addiction. Psychological capital is a positive psychological state manifested by individuals in the process of growth and development, mainly including four aspects: hope, self-efficacy, resilience, and optimism [26]. It was found that physical activity enhances the psychological capital of individuals. Stewart found that physical activity enhances self-esteem, self-confidence, and resilience [27]. Luthans also confirmed that physical exercise improves self-efficacy, belonging, and achievement. Furthermore, psychological capital reserves improved with physical exercise, and then a durable and relatively stable state of well-being was obtained [28]. In addition, high psychological capital should help college students use the mobile Internet correctly and reduce addiction. Simsek and Sali found a negative relationship between psychological capital and Internet addiction [29]. BI studies found that psychological capital directly helps ameliorate college students’ Internet addiction [30]. As one of the Internet terminals, smartphone addiction includes some attributes and characteristics of Internet addiction [31], which can be regarded as a new type of Internet addiction. In addition, Wu reported that low self-efficacy is a potential risk factor for Chinese smartphone users’ tendency to become addicted to social networking sites [32]. Therefore, we hypothesized that physical activity indirectly affects college students’ mobile phone addiction through the mediating effect of psychological capital.

Individuals with high levels of physical activity are likely to have better social adaptation. Social adaptation refers to the degree to which individuals adapt to their environment [33]. Chen and Hu described social adaptation as an adaptive capacity where people undergo various psychological, physiological, and behavioral adaptations to achieve a state of harmony with society to survive better; it also includes the subject’s ability to adjust their behavior to adapt to interpersonal interactions, such as social skills, handling skills, and interpersonal skills [34,35]. Some studies showed that after 12 weeks of aerobic exercise intervention, individuals’ ability to adapt to society improved [36]. Regular physical activity strengthens physical fitness, builds personality, and enables adolescents to develop interpersonal communication skills and enrich coping and problem-solving skills and techniques. Consequently, individuals’ overall social, coping, interpersonal, and other social adaptation skills improve, reducing external problem behaviors [37]. The social adaptation and comprehensive, adaptive development theories believe that the development of human social adaptability is based on continuous learning and exploration of life practice [38]. From this perspective, physical activity, as a type of positive social interaction activity, creates opportunities and platforms for interpersonal and social interactions for adolescents, enabling individuals to exchange emotions, share happiness and joy with their peers, establish a broader network of interpersonal relationships during exercise practice [39], and enhance their social adaptation ability.

In addition, empirical studies showed that improving and developing the social adaptability of children and adolescents is crucial to their future social growth, personality improvement, and academic adaptability; poor social adaptability may lead to communication difficulties, social withdrawal, social disorders, and a series of problem behaviors such as Internet addiction and aggression [40]. Inadequate adaptation can lead to feelings of alienation and social sensitivity disorders, disrupting everyday living and triggering Internet addiction disorder, mobile phone addiction, and other problem behaviors that seriously affect adolescents’ physical and mental development [41,42]. According to the behaviorist theory, individuals who are rewarded for certain behaviors are more likely to respond to such behaviors. Individuals with low social adaptability feel more frustrated and fail in reality. By contrast, individuals with negative emotions and dissatisfaction toward life temporarily escape their troubles through mobile phones and experience a sense of pleasure and relaxation by playing online games, socializing online, and accessing videos and music from their mobile network. For these reasons, their frequent use of mobile phones leads to mobile phone addiction [43]. Therefore, we hypothesized that physical activity indirectly affects college students’ mobile phone addiction through the mediating effect of social adaptation.

Psychological capital was also shown to be a protective factor for social adjustment in a few studies conducted on students’ psychological capital. One of the studies found that psychological capital mediated between negative life events and adaptation, and psychological capital was significantly and positively correlated with adaptation in college students [44]. High levels of psychological capital help individuals better integrate and adapt to society by regulating cognitive processes and producing positive behavior, enhancing an individual’s initiative and creativity to cope effectively with various work and life difficulties [45]. It was also found that hope is positively related to individual adjustment [46]. Another study reported that optimism predicts better adaptability among college students [47]. Relevant studies demonstrated that the development of psychological resilience facilitates individuals’ social adaptation and psychological well-being [48]. Xie’s research found that adolescents’ self-efficacy significantly predicts social adaptation and that adolescents with high self-efficacy experienced fewer negative emotional states such as anxiety or depression, coped strategically with changes in their surroundings, and maintained their physical and mental balance to adapt positively and confidently to their school and social environments [49]. Lower self-efficacy levels in college students can lead to worse academic and college adaptation [50]. According to the psychological capital buffer effect model, it is believed that psychological capital indirectly affects the state and behavior of individuals by affecting some intermediate variables [51]. Therefore, we hypothesized that psychological capital has a significant predictive effect on social adaptation and that physical activity can affect college students’ mobile phone addiction through the chain mediating effects of psychological capital and social adaptation.

In summary, we intended to systematically investigate the effects of physical activity, psychological capital, and social adaptation on university students’ mobile phone addiction, as well as the mediating role of psychological capital and social adaptation between physical activity and college students’ mobile phone addiction. Finally, we presented our results regarding mechanisms by which physical activity affects college students’ mobile phone addiction.

## 2. Materials and Methods

### 2.1. Participants

We conducted a cross-sectional study from March to April 2022 in Jiangsu Province, located in the eastern region of China. The research object included university students from Jiangsu Province, China. We selected 35 undergraduate schools for the survey through a preliminary study and research based on the type of school (public, private), quality of school (double-class universities, ordinary universities), and location of school (northern Jiangsu, central Jiangsu, southern Jiangsu, Nanjing) as the selection criteria. Data were obtained from 9406 students aged 16–29 years (M = 19.58, SD = 1.07) from freshmen to seniors, excluding postgraduate students. Participants accessed an electronic questionnaire through a QR code scanned by a mobile phone. A total of 10,863 questionnaires were collected. In order to ensure that the data collected were rigorous and valid, the questionnaires were screened and sorted to exclude those that were completed too early, randomly answered, irregularly filled, or incompletely answered. A total of 9406 valid questionnaires were collected with a valid recovery rate of 87%. The study protocol was approved by the Ethics Committee of the Medical College of Yangzhou University (YXYLL-2022-111).

### 2.2. Instruments

#### 2.2.1. International Physical Activity Questionnaire (IPAQ-L)

We used the International Physical Activity Questionnaire (IPAQ), developed by the International Physical Activity Task Force [52]. The long form of the Chinese version of the International Physical Activity Questionnaire used in this study applies to adults aged 15–69 and is divided into four main categories: study and work, transport, housework, and leisure activities, as well as including time spent sitting still, and is the questionnaire that is now widely used internationally to measure physical activity. This questionnaire assessed and calculated three intensity levels of PA, including low-intensity activity, moderate-intensity activity, and high-intensity activity. MET scores were calculated by multiplying the MET value for each activity by the duration (minutes) and frequency (days). The sum of these three activities is required to obtain the weekly physical activity. The questionnaire was proven to have good reliability and validity [53].

#### 2.2.2. Positive Psycho Capital Scale

A localized version in Chinese of the Positive Psychological Capital Questionnaire [54] developed by Zhang Kuo was used to measure the psychological capital of university students. The 26-item scale contains four factors: self-efficacy, hope, resilience, and optimism. Research showed that the power of co-motivation is more pronounced when psychological capital is considered a holistic concept (a higher-order factor) [55]. Scores on all items in the scale were summed, and a higher score indicated higher psychological capital. Cronbach’s alpha coefficient for this scale in this study was 0.926.

#### 2.2.3. Social Adaptation Diagnostic Questionnaire

The “Social Adaptability Diagnosis Scale” [34] compiled by Zheng Richang was used. The scale consists of 20 items, including 5 dimensions, and expresses social adaptability in the form of a total score, with higher scores indicating stronger adjustment ability and vice versa. In this study, Cronbach’s alpha coefficient for this scale was 0.753.

#### 2.2.4. Mobile Phone Addiction Scale

The Mobile Phone Addiction Scale, revised by Leung et al., was used [56]. The scale consists of 17 questions, with higher scores associated with higher levels of mobile phone addiction, including four factors: loss of control, withdrawal, avoidance, and ineffectiveness. In addition, questions 3, 4, 5, 6, 8, 9, 14, and 15 were used as screening questions for mobile phone addiction. In this study, Cronbach’s alpha coefficient for this scale was 0.938.

### 2.3. Data Analyses

Raw data were obtained from the Questionnaire Star questionnaire platform (https://www.wjx.cn/, accessed on 20 April 2022. SPSS 26.0 for Windows (IBM, Armonk, NY, USA) was used for statistical analysis. Descriptive analysis was used to describe the demographic characteristics of the participants. Enumeration data were described as means, standard deviations (SD), and percentages. In addition, gender differences in age, total PA MET, psychological capital scores, and social adaptation scores were investigated using *t*-tests. Spearman correlations were used to calculate correlations between each pair of metrics. Hayes’ PROCESS macro in SPSS (version 3.3, IBM, Armonk, NY, USA) was performed in the mediation analysis [57]. The bootstrap method was used to examine the significance of the mediating effect of psychological capital and social adaptation. We bootstrapped 5000 samples from the data and calculated the 95% bootstrap confidence intervals (CI). The mediating effect was considered significant if the CI did not contain 0.

## 3. Results

### 3.1. Control and Inspection of Common Method Deviation

Since we only used subjects’ self-reported answers to collect data in this study, the results may be affected by common method bias. Harman’s one-way test was used to examine common method bias. Based on the unrotated factor analysis results, a total of nine factors with a characteristic root >1 were extracted, and the explained variance of the largest factor was 21.73%, well below the critical criterion of 40%. Therefore, there was no significant common method bias in this study.

### 3.2. Participant Characteristics

A total of 9406 university students participated in this study. The characteristics of the participants are shown in Table 1. Male students reported more physical activity, higher psychological capital and social adaptation, and lower mobile phone addiction than female students. Pearson correlation coefficients between the variables were analyzed prior to analyzing the relationship between physical activity, psychological capital, social adaptation, and mobile phone addiction. Table 2 shows the correlations between the factors. Physical activity was significantly positively correlated with psychological capital and social adaptation and significantly negatively correlated with mobile phone addiction. Psychological capital was significantly positively correlated with social adaptation and significantly negatively correlated with mobile phone addiction. Social adaptation was significantly negatively correlated with mobile phone addiction.

### 3.3. Mediation Test of Psychological Capital and Social Adaptation

The results of the correlation analysis met the statistical requirements for further testing of mediating effects on psychological capital and social adaptation [58]. Bootstrap-based tests for mediating effects were performed using the SPSS macro program developed by Hayes, using Model 6, which was specifically designed to perform chained mediating model tests with gender, ethnicity, age, and grade as control variables; physical activity as the independent variable; psychological capital and social adaptation as mediating variables; and mobile phone addiction as the dependent variable for analysis.

The regression analysis results (Table 3) showed that physical activity significantly negatively predicted mobile phone addiction among college students (*β* = −0.049, *p* < 0.001). After including psychological capital and social adaptation in the regression equation, physical activity significantly positively predicted psychological capital (*β* = 0.1, *p* < 0.001) and significantly predicted social adaptation (*β* = 0.047, *p* < 0.001). Psychological capital significantly positively predicted social adaptation (*β* = 0.537, *p* < 0.001) but not mobile phone addiction (*β* = −0.021, *p* = 0.068). Social adaptation significantly negatively predicted mobile phone addiction (*β* = −0.256, *p* < 0.001), at which point physical activity still significantly negatively predicted mobile phone addiction (*β* = −0.02, *p* < 0.05).

The quantitative analysis results of the mediating effect showed (as shown in Figure 1 and Table 4) that psychological capital social adaptation played a significant mediating role between physical activity and mobile phone addiction among college students with a total mediating effect value of −0.028, accounting for 58.33% of the total effect of physical activity on mobile phone addiction—(effect value −0.048). The mediating effect specifically comprised indirect effects from two pathways: Profile effect 2 (effect value −0.012) from the physical activity → social adaptation → mobile phone addiction pathway, and indirect effect 3 (effect value −0.014) from the physical activity → psychological capital → social adaptation → mobile phone addiction pathway. Indirect effects 2 and 3 accounted for 25 and 29.2% of the total effects, respectively. The 95% confidence intervals for the aforementioned indirect effects did not include 0, indicating that both indirect effects reached a significant level.

## 4. Discussion

We investigated and analyzed college students’ physical activity and mobile phone addiction. Then, we discussed the internal relationship between physical activity and mobile phone addiction in college students at this stage. In this study, college students scored 43.08 ± 13.86 for cell phone addiction, and 35.9% of college students showed symptoms of cell phone addiction. A survey showed that the detection rate of cell phone addiction among college students was 21.4–27.4% [13], indicating that with the development of the times, mobile phones have become an important tool in the daily life of college students, causing college students to increasingly rely on them and resulting in severe mobile phone addiction.

This study found that female college students had significantly higher scores for mobile phone addiction than male college students, confirming previous research [23]. According to Jiang and Demirci [59,60], compared with male students, female college students use mobile phones more frequently for interpersonal communication, entertainment, and online shopping and tend to communicate with others through mobile phones to establish and maintain their social relations. These differences account for the significantly higher degree of mobile phone addiction in female college students than in male students.

The regression analysis results that controlled for gender, ethnicity, age, and grade show that physical activity negatively predicts the mobile phone addiction of college students; that is, the more physical activity college students perform, the less likely they are to become addicted to mobile phones, which is consistent with previous studies [23]. Adolescents with severe mobile phone addiction habitually ignore the surrounding environment and real-world interpersonal communication, easily indulge in the virtual network world mediated by mobile phones, and increase static screen behaviors, affecting the level and experience of daily physical activities at school [61]. Increased physical activities, especially leisure-related sports, cause the pituitary gland to secrete endorphins, which compete with addictive substances in the central nervous system for receptors and induce happiness, thus inhibiting addiction [62]. In addition, physical activity can also reduce the amount of time individuals spend on mobile phones and their dependence on social networks [63], thereby reducing the possibility of mobile phone addiction.

Our findings show that physical activity impacts mobile phone addiction among university students through the separate mediating role of social adaptation. One study found that individuals who engage in active leisure activities (walking, gym workouts, cycling, etc.) reported higher social adjustment scores [64]. College students’ social adaptation ability mainly refers to their adaptation to studying, interpersonal relationships, career selection, university life, and emotions, which are expressed by self-control, self-confidence, strong interpersonal abilities, and emotional regulation during the adaptation process. The social adaptation and integrated adaptive development theories suggest that the development of human social adaptability is based on continuous learning and the exploration of life practices [39]. From this level of understanding, physical activity encompasses outdoor and leisure activities. These activities, as a positive social interaction, create a platform for interpersonal and social interactions, enabling individuals to communicate positively during physical activity and establish a broader network of interpersonal relationships [40], bringing psychological fulfillment, enriching university life, and enhancing social adaptation skills.

Secondly, if adolescents are fulfilled at home and in society, they will exhibit healthy behaviors of using the Internet with moderation and experiencing positive emotions [65]. Davis’ research found that weak social skills and non-adaptive cognitions are prerequisites for problematic behavior, implying that social adaptability can determine the choice and expression of an individual’s social behavior [66]. Evidence shows that social adaptation deficits can lead to other problematic behaviors such as internet addiction and mobile phone addiction [42,43]. Therefore, it is reasonable to assume that individuals with good social adjustment skills may spend more time in real life making new friends and experience less of the negative, frustrating feeling of not being able to adapt to society. The use of mobile phones to fill the boredom and emptiness in your life is then significantly reduced, thus reducing your dependence on them.

Furthermore, some studies showed that psychological resilience protects people from addictive behaviors [67], and self-efficacy provides individuals with better self-control [68]. These positive psychological traits can lead to better self-control and prevent addictive behavior. However, our findings showed that the mediating role of psychological capital in physical activity and mobile phone addiction was not significant and psychological capital did not significantly predict mobile phone addiction, which is consistent with the findings of previous studies [69]. One possible reason is that other mediating variables must influence psychological capital to impact mobile phone addiction. The psychological buffer effect model states [51] that psychological capital may indirectly affect outcome variables at the individual, group, and organizational levels by influencing several intermediate variables. The intermediate variables presented here contain mediating and moderating variables, implying that the effect of psychological capital on the outcome variable may be indirect rather than direct.

Our findings show that physical activity impacted mobile phone addiction through the chain mediating effects of psychological capital and social adaptation. Among these, psychological capital significantly and positively predicted social adaptation. The research found that individuals’ psychological capital influences their positive emotions, directly affecting their attitudes and behaviors [70] and contributing to their strengths in adaptive and prosocial behavior [71]. Psychological resource theory considers psychological capital as a critical psychological resource for individuals and one of the important indicators of their psychologically healthy development, which effectively mobilizes and manages other resources to improve social adjustment [23]. It is also an important protective resource for development [72]. Psychological capital, as a collection of positive psychological states, positively influences individuals’ attitudes, emotions, and behaviors; the role of these positive emotions and feelings in an individual’s social adjustment is evident [73]. Higher levels of physical activity in university students improve their self-efficacy, mental toughness, etc., in a wide range of physical activities [74,75], thus enhancing their psychological capital. The higher the psychological capital of university students, the better their social adjustment, thereby improving their interpersonal relationships and reducing mobile phone addiction.

Although these positive findings support this paper’s findings, our study is not without its limitations. First, our study was cross-sectional with a predictive result that failed to reveal the causal relationships involved. Second, we measured physical activity using the self-report questionnaire IPAQ, and the measured results were subjective. Therefore, future research must improve the research paradigm and methods by combining observation, measurement, experimentation, and tracking methods with the help of advanced scientific and technological tools. Further research should consider using more objective measurement devices (e.g., accelerometers and pedometers) to measure physical activity levels, delve deeper into the mechanisms that influence the mental activity of university students, and scientifically and rationally explain the cause-and-effect relationship.

## 5. Conclusions

The results of our study indicate that physical activity significantly and negatively predicted college students’ cell phone addiction. Social adaptation mediated between physical activity and college students’ mobile phone addiction, whereas psychological capital did not. Social adaptation and psychological capital mediated the chain between physical activity and mobile phone addiction among college students. After combining the study findings, we suggest that colleges should create an excellent interpersonal atmosphere on campus and improve the social adaptability of college students through diverse interpersonal interactions, which can effectively alleviate or avoid mobile phone addiction, thus helping college students form positive and active physical activity behaviors. At the same time, colleges should further improve the physical education curriculum system and the construction of extracurricular sports organizations (societies/clubs) and other institutions, accelerate the development of campus sports culture, and create a positive and healthy sports atmosphere, so that college students have more time and opportunities to engage in physically actively and mentally beneficial activities and avoid the tendency to develop mobile phone addiction.

## Figures and Tables

**Figure 1 ijerph-19-09286-f001:**
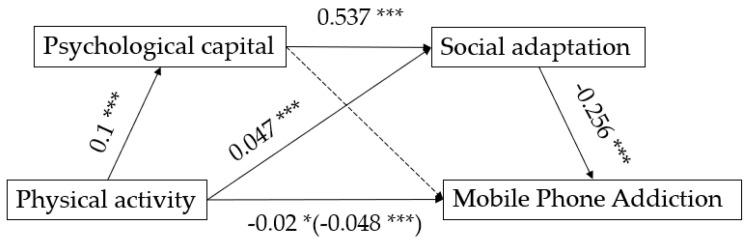
Mediation model of psychological capital and social adaptation between physical activity and mobile phone addiction. Note: * means that *p*-value is <0.05; *** means that *p*-value is <0.001.

**Table 1 ijerph-19-09286-t001:** Participants’ demographics and characteristics.

Participant Characteristics	Men (*n* = 3516)		Women (*n* = 5890)		Total (*n* = 9406)		
	Mean or *n*	±SD or %	Mean or *n*	±SD or %	Mean or *n*	±SD or %	*p*-Value
**Age (years)**	19.69	1.1	19.52	1.04	19.5	1.07	<0.001
**Nation**							
Han Chinese	3357	95.50%	5574	94.60%	8931	95%	<0.001
Others	159	4.50%	316	5.40%	475	5%	
**Grade**							
First grade	1803	51.30%	2918	49.50%	4721	50.20%	
Second grade	1611	45.80%	2727	46.30%	4338	46.10%	
Third grade	79	2.20%	198	3.40%	277	2.90%	
Fourth grade	23	0.70%	47	0.80%	70	0.70%	
**Physical activity**							
Total MET minutes/week	3485.01	3102.46	2588.13	2548.89	2923.39	2802.44	<0.001
**Psychological Capital**							
PPQ (score)	120.58	22.7	118.63	19.33	119.36	20.67	<0.001
**Social adaptation**							
SAFS (score)	3.79	11.28	1.52	11.79	2.37	11.65	<0.001
**Mobile phone addiction**							
MPAI (score)	41.92	14.96	43.78	13.12	43.08	13.86	<0.001
Addiction	1279	36.4%	2237	35.6%	3377	35.9%	
Not addictive	2098	63.6%	3792	64.4%	6029	64.1%	

**Table 2 ijerph-19-09286-t002:** Correlation between variables.

	Physical Activity	Psychological Capital	Social Adaptation	Mobile Phone Addiction
Physical Activity	1			
Psychological Capital	0.106 **	1		
Social Adaptation	0.113 **	0.544 **	1	
Mobile Phone Addiction	−0.06 **	−0.167 **	−0.275 **	1

Note: *n* = 9406 ** *p* < 0.01.

**Table 3 ijerph-19-09286-t003:** Regression analysis of variable relationships in the model.

Regression Equation		Overall Fit Index			Significance of Regression Coefficient		
Result Variable	Predictive Variable	R	R^2^	F	*β*	SE	t
Mobile phone addiction	Gender	0.09	0.01	15.233	0.121	0.022	5.589 ***
	Ethnicity				0.129	0.047	2.737 **
	Age				0.023	0.012	1.906
	Grade				0.003	0.022	0.141
	Physical activity				−0.048	0.01	−4.739 **
Psychological capital	Gender	0.13	0.02	31.681	−0.066	0.021	−3.071 **
	Ethnicity				−0.159	0.047	−3.402 ***
	Age				−0.037	0.012	−3.12 **
	Grade				−0.039	0.021	−1.844
	Physical activity				0.1	0.104	9.686 ***
Social adaptation	Gender	0.55	0.3	682.359	−0.131	0.018	−7.233
	Ethnicity				0.044	0.039	1.131
	Age				−0.005	0.01	−0.546
	Grade				0.04	0.018	2.216 *
	Physical activity				0.047	0.009	5.421 ***
	Psychological capital				0.537	0.009	61.902 ***
Mobile phone addiction	Gender	0.28	0.08	115.172	0.077	0.02	3.688 ***
	Ethnicity				0.114	0.045	2.524 *
	Age				0.015	0.011	1.347
	Grade				0.006	0.02	0.327
	Physical activity				−0.02	0.01	−2.023 *
	Psychological capital				−0.021	0.012	−1.819
	Social adaptation				−0.256	0.012	−21.66 ***

Note: gender, nationality, age, and grade all adopt virtual coding. * means that *p*-value is <0.05; ** means that *p*-value is <0.01; *** means that *p*-value is <0.001.

**Table 4 ijerph-19-09286-t004:** Chain-mediated model effect tests for psychological capital and social adaptation.

Benefit Type	Effect Value	BootSE	Bootstrap 95% CI		Proportion of Relative Effect
			Boot LLCI	Boot ULCI	
Total effect	−0.048	0.01	−0.0691	−0.0283	100%
Direct effect	−0.02	0.01	−0.0402	−0.0006	41.67%
Indirect effect 1	−0.002	0.002	−0.0053	0.0008	4.20%
Indirect effect 2	−0.012	0.002	−0.0169	−0.0076	25%
Indirect effect 3	−0.014	0.002	−0.0174	−0.0107	29.20%
Total indirect effect	−0.028	0.003	−0.0344	−0.0225	58.33%

Note: Boot SE, Boot LLCI, and Boot ULCI refer to the standard errors and lower and upper 95% confidence intervals of the indirect effects estimated by the bias-corrected percentile Bootstrap method, respectively. Indirect effect 1: physical activity → psychological capital → mobile phone addiction; indirect effect 2: physical activity → social adaptation → mobile phone addiction; indirect effect 3: physical activity → psychological capital → social adaptation → mobile phone addiction.

## Data Availability

The data presented are available on request from the corresponding author.

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
