# Peer review of "The Relationship between Physical Activity and College Students’ Mobile Phone Addiction: The Chain-Based Mediating Role of Psychological Capital and Social Adaptation"

_ijerph, 2022, doi:10.3390/ijerph19159286_

Round 1
Reviewer 1 Report
The paper is very well-founded and written, contains robust statistical Analysis and a large number of participants, which makes it excelente. Small modification are necesary to improve the manuscript.
1. Could have a greater connection between the paragraphs of the introduction. There are many concepts described, but there is a connection between them, which could be more explicit in the text.
2. I suggest Adding the IPAQ validity evidence values to be standardized with the Other instruments used.
Author Response
Response to Reviewer 1 Comments
Response: Thank you very much.
Point 1: Could have a greater connection between the paragraphs of the introduction. There are many concepts described, but there is a connection between them, which could be more explicit in the text.
Response 1: Agreed. We have added transitional sentences between paragraphs to make the transition gentle(Pg2, L71-72; L90-91).
Point 2: I suggest Adding the IPAQ validity evidence values to be standardized with the Other instruments used.
Response 2: Thank you for your significant reminding. We provide evidence of the reliability of the IPAQ (Pg4, L186-187).
Reviewer 2 Report
It is necessary to improve the conclusions, an important study with a good approach cannot have such limited conclusions.
It would be interesting to propose future research on the use of the mobile phone, in this study it is considered as a generic use but it can be used for different purposes. In addition to addiction, it is convenient to know what it is for the person.
I consider it important that, in addition to the three conclusions reached by the authors, some line of future research is proposed or some hypothesis of the authors about the relevance of their conclusions. Although this was obviously not among the objectives of the study, it seems interesting that the authors could write it, it would open up interesting lines of research.
Author Response
Response to Reviewer 2 Comments
Response: Thank you very much for your previous comments that helped us improve this manuscript.
Point 1: It is necessary to improve the conclusions, an important study with a good approach cannot have such limited conclusions.
Response 1: Your question is very reasonable. We have added to the conclusion section and added recommendations related to how colleges can promote physical activity, social adaptation and reduce mobile phone addiction among college students (Pg10, L386-L399).
Point 2: It would be interesting to propose future research on the use of the mobile phone, in this study it is considered as a generic use but it can be used for different purposes. In addition to addiction, it is convenient to know what it is for the person.
Response 2: Thank you very much for your advice. We have only considered the negative effects of mobile phone use on people and ignored the effects it has on them in other ways. Your ideas are interesting and we will consider your suggestions in our follow-up study.
Point 3: I consider it important that, in addition to the three conclusions reached by the authors, some line of future research is proposed or some hypothesis of the authors about the relevance of their conclusions. Although this was obviously not among the objectives of the study, it seems interesting that the authors could write it, it would open up interesting lines of research.
Response 3: Thank you very much for your approval. Our group has begun further research into the psychological mechanisms by which physical activity reduces mobile phone addiction among collage students, with the aim of fully revealing how physical activity reduces mobile phone addiction among collage students.
Reviewer 3 Report
The study is particularly relevant for the present day times, when addictions and other health-related problems are unfortunately increasingly common among young people, and solutions need to be found in this respect. The study proves once again, on an impressive sample, that handy physical activities represent a solution that needs to be considered.
The main conclusions for the present article are the following:
· - the title and key words comply with the content of the article;
· - the information is synthetized and clearly and logically presented;
· - information about the research design and the results are clearly provided
· - the references used are in compliance with the topic and relevant.
Comments and suggestions for authors:
- the content presented should be contextualized a bit more, by making short reference to peculiar features of the Chinese people with respect to practising sports, mobile phone use, ways of socializing and, then, possibly, by connecting these data to some data worldwide
- as the data were collected in 2022, it would be worth considering the possible impact of the pandemic on the results
- some proposals could be included with reference to actions that university management could take to facilitate students' physical activity during school
- there are some slight slips in point of the use of the English language. they do not influence the overall quality of the paper, but I suggest the authors should remedy them:
L2: “Physical Activity on” – Physical Activity and ….
L3: “Mediation Role” - Mediating Role
L21 – “and indirectly affects” – and it indirectly affects
L27 – no ;
L88 – “individual’s’ ability”- individuals’ ability
L182, 187, 193 – “The Cronbach’s alpha” - “Cronbach’s alpha”
L318 – “adolescents can be” - “adolescents are”
L319 – “Internet in moderation” - “Internet with moderation”
Author Response
Response to Reviewer 3 Comments
Response: Thank you for your comments. We have gone through your comments carefully and tried our best to address them one by one.
Point 1: the content presented should be contextualized a bit more, by making short reference to peculiar features of the Chinese people with respect to practising sports, mobile phone use, ways of socializing and, then, possibly, by connecting these data to some data worldwide
Response 1: Your question is very reasonable. We have added relevant references based on your suggestions (Pg1, 37-38; Pg2, 47-50; 53-54).
Point 2: as the data were collected in 2022, it would be worth considering the possible impact of the pandemic on the results
Response 2: Your question is very reasonable. However, thanks to the Chinese government's prevention policy, the pandemic was contained in 2020 and there were no mass infections in 2022, and society in the areas we surveyed was functioning normally and the lives of collage students were not affected by the pandemic. Therefore, we believe that the results of this study are not influenced by the pandemic variable.
Point 3: some proposals could be included with reference to actions that university management could take to facilitate students' physical activity during school
Response 3: Agreed. We have added recommendations to promote physical activity among college students during school in the conclusion section (Pg10, Pg10, L386-L399).
Point 4: there are some slight slips in point of the use of the English language. they do not influence the overall quality of the paper, but I suggest the authors should remedy them.
Response 4: Thank you very much for discovering this error. According to your suggestion, we corrected the above grammatical errors and checked the rest again. We have sent the paper to MDPI for editing before submission and apologize for any errors that may have occurred due to subsequent revisions.